# Study of Titanium–Silver Monolayer and Multilayer Films for Protective Applications in Biomedical Devices

**DOI:** 10.3390/molecules26164813

**Published:** 2021-08-09

**Authors:** Sebastián Mina-Aponzá, Sandra Patricia Castro-Narváez, Luz Dary Caicedo-Bejarano, Franklin Bermeo-Acosta

**Affiliations:** 1Faculty of Basic Sciences, Campus Pampalinda, Universidad Santiago de Cali, Cali 760035, Colombia; sebastian.mina00@usc.edu.co (S.M.-A.); ludcaice@usc.edu.co (L.D.C.-B.); frank@usc.edu.co (F.B.-A.); 2Electrochemistry and Environment Research Group (GIEMA), Universidad Santiago de Cali, Cali 760035, Colombia; 3Mycology Research Group (GIM), Universidad Santiago de Cali, Cali 760035, Colombia; 4Physics Statistics and Mathematics Research Group (GIFEM), Universidad Santiago de Cali, Cali 760035, Colombia

**Keywords:** corrosion, magnetron sputtering, wettability, antimicrobial activity

## Abstract

The search for coatings that extend the useful life of biomedical devices has been of great interest, and titanium has been of great relevance due to its innocuousness and low reactivity. This study contributes to the investigation of Ti/Ag films in different configurations (monolayer and multilayer) deposited by magnetron sputtering. The sessile droplet technique was applied to study wettability; greater film penetrability was obtained when Ag is the external layer, conferring high efficiency in cell adhesion. The morphological properties were characterized by SEM, which showed porous nuclei on the surface in the Ag coating and crystals embedded in the Ti film. The structural properties were studied by XRD, revealing the presence of TiO_2_ in the anatase crystalline phase in a proportion of 49.9% and the formation of a silver cubic network centered on the faces. Tafel polarization curves demonstrated improvements in the corrosion current densities of Ag/Ti/Ag/Ti/Ag/Ti/Ag/Ti and Ti/Ag compared to the Ag coating, with values of 0.1749, 0.4802, and 2.044 nA.m^−2^, respectively. Antimicrobial activity was evaluated against the bacteria *Pseudomonas aeruginosa* and *Bacillus subtilis* and the yeasts *Candida krusei* and *Candida albicans,* revealing that the Ti/Ag and Ag/Ti/Ag/Ti/Ag/Ti/Ag/Ti coatings exhibit promise in biomedical material applications.

## 1. Introduction

The studies and fabrication of biomaterials have advanced in the last few decades due to their great demand in clinical applications and joint implants, since they provide high biocompatibility with the host organism, can efficiently replace certain functions, do not produce corrosion, are easy to adapt, are reproducible, and present low costs in their fabrication [1].

In particular, titanium has been widely used in prostheses due to its harmlessness, low reactivity and good osseointegration (structural and functional connection between living bone and an implant surface) [2,3]. However, its high cost has led to the use of materials such as metallic alloys [4], ceramics [5], and polymers [6] that can be affected by the characteristics of the medium in which the implants are made by presenting microbial incompatibilities [7] and changes due to corrosion [8]. The search for coatings that extend an implant’s useful life has become necessary and indispensable, and it has set a trend in materials studies [9]. Previous studies have shown that Ag has antimicrobial characteristics and is highly stable in the presence of titanium films [10].

Titanium is used in air motors, aerospace structures, capacitors, valves, hip implants, fasteners, and dental casings due to its corrosion resistance characteristics and high mechanical stability [11]. This metal does not show antibacterial properties by itself, so it has become essential to integrate active bactericidal coatings when it is used as a biomaterial to prevent adhesion, colonization, and bacterial infection in implants, as these can lead to implant failure that requires surgical intervention and a supply of antibiotics [12]. Microbial resistance to antibiotics and antifungals is currently considered alarming by the World Health Organization (WHO) and resistance to multiple drugs is a challenging problem in medical care worldwide, so numerous investigations have positioned Ag as an effective agent [13,14].

The surfaces of biomedical devices are exposed to corrosion at the interface of the metallic material and body fluids due to the high concentration of chloride ions in blood plasma (113 mEq.L^−1^), as well as the presence of amino acids and proteins that accelerate this process. Titanium-based coatings or alloys are of great interest due to the fact that a passive titanium oxide film is present on their surface (it spontaneously grows after exposure to air), providing favorable properties that can protect the metal from further oxidation, as well as enable stability at different pH ranges [15]. Due to the exceptional performance of TiO_2_, multiple researchers have focused on its formation in different allotropic forms such as anatase, rutile, and brookite. The brookite form consists of six oxygen atoms coordinated with a titanium atom in a distorted octahedral configuration, and the rutile and anatase phases have significant value due to their excellent properties in titanium coatings that have led to them finding various applications at the industrial level [16].

At present, there has been a considerable amount of research on the reduction of bacterial growth using in vitro studies, where favorable results have been obtained by incorporating metals such as Cu^2+^, Zn^2+^, and Ag^2+^ as coatings in implants [17]. Silver has been the subject of growing interest because its ions bind to thiol groups present in the proteins of many microorganisms, thus allowing it to alter the structure of and thus rupture proteins in the bacterial cell wall; it has also been shown to alter several crucial enzymes in cellular respiration and metabolism, thus causing bacterial inhibition. Silver nanoparticles evaluated against bacteria (Gram-positive and Gram-negative), fungi, algae and viruses suggest antimicrobial properties [18,19,20].

Metals exhibit the ability to alloy with emerging intermetallic compounds and to selectively improve their properties by increasing stiffness, increasing corrosion resistance, and decreasing weight with their mixed chemical character [21]. The titanium–silver alloy presents an excellent combination of the properties of both metals, with silver providing an increase in biocompatibility because it prevents the formation of bacterial biofilms that would lead to the generation of inflammation or infection around an implanted device—a crucial parameter in implants and prostheses [22].

Biomaterials are materials used in devices for clinical use that are designed to replace specific parts of living systems. They are characterized by their interactions with tissue, blood, or body fluids, so they must adopt an inert position in front of the host organism while still fulfilling their established function; such a function ranges from replacing soft tissues (blood vessels or artificial skin) to replacing hard tissues (bone or joint implants, artificial dental implants, cardiac pacemakers, or biosensors) [23]. Their stability is determined by the surface properties of the implanted coating, such as the chemical composition of the alloys, surface roughness, and the implementation of pretreatments [24].

Despite efforts to ensure sterility, implantable and prosthetic medical devices are easily contaminated when incorporated into the body, thus leading to secondary surgical procedures such as replacement, amputation, or mortality. Some sources of infectious agents are the local environment (such as the air and surfaces, surgical equipment, and medical staff clothing in the operating room) and the bacteria present on the patient’s skin [25]. The most frequent pathogens of bacterial-infection-forming biofilms on medical devices are *Staphylococcus aureus, Staphylococcus epidermidis, Streptococcus viridans, Proteus mirabilis, Klebsiella pneumoniae, Bacillus subtilis, Enterococcus faecalis, E. coli*, and *Pseudomonas aeruginosa* [26]. Pathogenic fungi mainly include species of *Candida* spp. *(C. albicans, C. glabrata, C. parapsilosis, C. tropicalis,* and *C. krusei), Aspergillus, Cryptococcus, Trichosporon, saccharomyces,* and *Fusarium* [27,28].

Incorporation of antibacterial agents via hybrid coatings, micro- and nanoscale modifications and biomimetic functionalization of titanium surfaces demonstrate rapid osseointegration [29,30]. The implantation of dual Zn/Ag ions in osteogenic activity presents synergistic effects on the antibacterial capacity of titanium due to the long-range interactions of Zn and the short-range interactions of Ag derived from microgalvanic pairs in co-implanted titanium [31]. The use of Ag nanocomposites modified by friction stir processing establishes an antibacterial effect independent of the release of Ag+ ions and the number of silver nanoparticles embedded on the surface, with no cytotoxicity to bone mesenchymal stem cells in vitro [32]. Also, the use of porous titanium implants by electrolytic oxidation of plasma with strontium and silver nanoparticles has demonstrated excellent surface biofunctionalization [33].

There has been enormous progress in the fabrication of thin films; several types of techniques have been established for coating different materials such as sputter deposition [34], physical vapor deposition techniques [35], sol–gel coatings [36], electrochemical methods [37], and electrospray [38]. The sputtering process is based on a (non-thermal) evaporation in which the atoms present in the surface of the material are ejected by momentum transfer, which is the result of bombarding the target with particles at an atomic-size energy level [39,40,41]. 

A liquid penetrates a solid when the contact angle is less than 90°. Penetrability or wetting is poor for materials with low surface energy; it has been reported that the surface energy values for metals, metal oxides, and ceramics are usually above 500 mJ.cm^−2^, which facilitates the interaction between the implant and biological medium [42]. The surface energy of the solid is proportional to the wettability determined by the contact angle between the surface and the physiological medium [43].

This project provides information on the surface energy of Ti and Ag thin films in different configurations (monolayer and multilayer) obtained by cation bombardment coating. It establishes microbiological adhesion of xxx and corrosion analysis in Ringer’s lactate salt solution. This information is relevant in biomaterials, especially for lifetime projections.

## 2. Analysis of Results

### 2.1. Characterization of Deposited Films

#### 2.1.1. Morphological Analysis

The Ti and multilayer films showed platinization with conductive character and high adhesion on the surface. The Ag coating (Figure 1a) exhibited the presence of crystals; it was prone to corrosion, as demonstrated in few porous nuclei on the surface with an average size of 16.98 µm, which made it prone to detachment; the observed roughness was a response to the oxidation process experienced on the surface of the film [44]. The Ti coating (Figure 1c) was homogeneous and uniform, without embedded crystals. In the enlarged image of Figure 1, some particles can be seen in the layer without structural changes. The morphologies of the monolayer and multilayer films are observable in Figure 1e–j. The Ti/Ag film had infiltrated crystals over the entire surface; in the enlarged view, a small proportion of pores of approximately 6.37 µm in size can be observed. The Ti/Ag/Ti/Ag coating presented a layer on the homogeneous surface, with a decrease in impregnated crystals randomly distributed on the surface of the substrate; layers marked by faint lines can be distinguished. Finally, the Ag/Ti/Ag/Ag/Ti/Ag/Ag/Ti coating presented three uniform and homogeneous layers that are perceptible due to the low adhesion that caused its detachment, an aspect that diminishes the viability of its protective properties against corrosion; there was evidence of embedded crystals and the formation of pores on the film.

#### 2.1.2. Structural Characterization of Films

Structural characterization by X-ray diffraction via the scratch beam technique revealed diffractograms of the thin film surface without detecting the crystalline phase of the substrate and/or the innermost layers of the coating (Figure 2). The Ti film exhibited diffraction peaks on the 2θ axis at 38°, 49°, 56°, 64°, 70°, and 76°, which was characteristic of the anatase phase (tetragonal crystalline system) of the TiO_2_ passivating layer formed on the surface with a composition of 49.9%. It was also possible to identify peaks at 2θ at 38°, 43°, 64°, 76°, and 80°, which corresponded to titanium oxide with a low crystallinity or in an amorphous state with a dominant composition of 50.1% in the XRD pattern [45,46]. The Ag film exhibited five 2θ peaks at 38°, 44°, 64°, 64°, 77°, and 81° that resembled the face-centered cubic lattice (FCC) peaks found in a study by Ma et al. [47].

The Ti/Ag bilayer coating showed five peaks that revealed the presence of Ag as the external coating of the film, which is in agreement with the results found for the silver film, but with weaker diffraction intensity. The Ti/Ag/Ti/Ag multilayer films revealed a composition of 66.1% silver and 33.9%, and the presence of pores and crystals embedded on the coating made it possible to detect Ti even when the top layer was Ag. Finally, the Ag/Ti/Ag/Ti/Ag/Ti/Ag/Ti coating showed the presence of Ag in the diffractogram, which may be due to the low adhesion of the layers evidenced in the SEM analysis, where three layers are observed.

The composition of the multilayer coatings can be explained by previously reported results that stated that in multilayer deposits between Ag and Ti, an interface of an AgTi intermetallic compound with a hard and brittle complex crystalline structure, characterized by covalent bonds and higher energy ionic bonding, is formed and prevents the individual species from completely mixing [48]. The above-mentioned results may explain the three Ag/TiAg/Ag layers in the Ag/Ti/Ag/Ag/Ti/Ag/Ag/Ti coating with poor adhesion at the Ag interface. In contrast, in the Ti/Ag/Ti/Ti/Ag coating, Ag layers were formed encapsulated between more stable Ti layers (Ti/Ag/Ti) imparting homogeneity to the coating.

#### 2.1.3. Wettability Analysis

Wetting results in a polar solvent (water), a strongly basic component (glycerol), an aprotic polar component (propylene carbonate) and a basic component (formamide) coincide with typical surface adhesion analyses. This study added an isotonic solution (Hartmann) for the purpose of establishing the conditions of the films when exposed to a medium with blood components. Figure 3 shows the contact angle of the water droplet with the Ti/Ag/Ti/Ag film when deposited on the surface and after 120 s.

The solvents in the different films presented contact angles of less than 90°, which is consistent with good penetrability and hydrophilic character (greater affinity for polar solvents) [49]. The contact angle decreased in the polar solvents and increased in the saline and aqueous solutions, thus showing a high compatibility for implant coatings. As the multilayers of the coatings increased, the angle increased in all the studied solvents (Figure 4).

Previous studies have concluded that increasing the contact angles results in a decrease in both wettability and surface energy, thus decreasing the cell adhesion of the coating [38]. The measurement of the contact angle is a determining factor of the wettability of a film. Through the SEM analysis of the morphology of the studied surfaces, a tendency justifying the penetrability of the coatings in layers and multilayers could be predicted. In the coatings where the outermost layer was Ag, a higher porosity was evidenced and allowed the wettability to increase. In contrast, when the outermost layer of the coating was Ti, a higher uniformity and lower wettability were evidenced, as supported by a study conducted by E. Matykina et al. [50] where the contact angle decreases with increasing surface roughness in hydrophilic and hydrophobic solvents. The Ag films presented deficiency in coating adhesion on the glass substrates, which could indicate that cohesion forces predominate over adhesion forces, resulting in films that are moderately brittle and susceptible to detachment upon interaction with a surface [51].

Table 1 summarizes the results of the surface energy analysis of the different coatings by implementing the Young–Dupre equation, which relates the contact angles to the energies in the liquid–vapor interface (γ_LV_), the angle formed in the solid–liquid interface (cosϕ_SL_), and the solid–liquid adhesion (W_SL_).
(W_SL_ = γ_LV_ (1 + cosϕ_SL_)(1)

X-ray diffraction analysis establishes that the multilayer Ag/Ti/Ag/Ag/Ag/Ti/Ag/Ti coating has low surface energy due to the outer Ti layer, which gives it low penetrability properties (higher contact angles) of the Hartmann solution. In the Ti/Ag/Ti/Ag film, the presence of both metals was detected due to the diffusion of titanium towards the outer layer of the coating as a result of the oxidation process exhibited by Ag, evidenced by a gradual increase in the decrease of the contact angles. In the Ti/Ag coating exhibited the highest wettability of the multilayer coatings; the Ti coating was not detectable in the X-ray diffractogram.

Therefore, the wettability properties of the coating were mainly influenced by the properties exhibited by the last layer. In all films, wettability increased with time; the contact angle was more reliable when the droplet was able to stop its oscillation or reach its maximum area (equilibrium). When the droplet diameter exceeds the capillary length, the droplet weight is a parameter that favors deformation [52].

#### 2.1.4. Corrosion Study of the Films

The anodic and cathodic Tafel polarization curves of each of the monolayered and multilayer coatings when exposed to Hartmann’s solution or Ringer’s lactate (without degassing because we desired to simulate body solutions) are shown in Figure 5. Physically, some surfaces of the area exposed to polarization formed a yellowish layer due to the oxidation process experienced in the immersion of Hartmann’s solution; this layer was more noticeable in the Ag coating.

The extrapolation of the Tafel curves shows that the multilayer coatings presented good corrosion resistance with respect to the Ag coating, which indicates that the film promoted the attack of the solution in which it is immersed due to the degree of porosities formed on its surface. As reported by Correa et al. [34], these porosities can be reproduced due to nucleation phenomena caused by Ti, which confers an increase in this property in the coatings of these alloyed metals due to an increase in microstructural homogenization and the presentation of a crystalline phase that can be observed by XRD analysis; these phenomena were previously reported by Kumari and Majumdar [53]. In the multilayer coatings, the intermetallic character was reflected by its intermediate corrosion current (i_corr_) values, to which Ag contributed an increase due to its ease of corrosion and presentation of little homogeneity or few pores on its surface.

Table 2 shows the results obtained for potential and current density, where it can be seen that the multilayer coating that showed the best properties was the Ag/Ti/Ag/Ti/Ag/Ti/Ag/Ti film, followed by Ti/Ag and Ti/Ag/Ti/Ag. The titanium monolayer coating showed excellent resistance to corrosion; its potential was −0.2791 V with a J_corr_ of 0.2899 nA.cm^−2^.

By decreasing the Ti/Ag layers, smaller i_corr_ values than those of the Ti coating (monolayer) were obtained. However, when the innermost layer was Ag, the anticorrosive properties of the film were increased, as in the case of the Ag/Ti/Ag/Ag/Ti/Ag/Ti coating, where the arrangement of the external Ti layer favored the uniform formation of a TiO_2_ layer (anatase crystallinity according to XRD results) on the surface that protected the metal from suffering additional oxidation. The coating showed an i_corr_ of 2.798 nA, thus making it more protective than that observed in the Ti film. Comparable results were presented by Cardenas [54].

### 2.2. Microbiological Analysis

The microbiological analysis results (in colony-forming units (CFU)) corresponding to the adherence of microorganisms evaluated on the surface of the coatings did not exceed 6.12% for yeasts and 27.08% for bacteria (Table 3). Figure 6 shows the CFU of *C. krusei* from samples taken from the different treatments after 96 h of incubation at room temperature. The lowest adherence was found in the TiAg-coated film exposed to *P. aeruginosa* bacteria, with 2.40 × 10^6^ CFU (0.17%), and *B. subtilis,* with 1.3 × 10^7^ CFU (1.08%), after incubation at 37 °C for 24 h in the soy trypticase agar culture medium. Yeast strains exhibited the lowest growth on Ag/Ti/Ag/Ag/Ti/Ag/Ti/Ag/Ti-coated film: *C. krusei* exhibited 1.3 × 10^7^ CFU (3.13%) and *C. albicans* exhibited 1.0 × 10^8^ CFU (0.15%) on Sabouraud agar after 48 h of incubation at room temperature.

Ag-dominated multilayer coatings are susceptible to microbiological adhesion. Lenis et al [49] reported that coatings with increased roughness and low crystallinity or amorphous state (as is the case of Ti/Ag/Ti/Ag) show a favorable response in terms of adhesion.

Three main mechanisms have been reported in regions where antibacterial attacks occur and are influenced by the action of released silver ions: (1) the alteration of vital cellular processes by binding to sulfhydryl or disulfide functional groups on the surfaces of membrane proteins and other enzymes, (2) DNA condensation as a defense mechanism that limits the ability of cells to self-replicate, and (3) oxidative stress caused by the catalysis of the formation of reactive oxygen species (ROS) [55]. The obtained results showed a greater inhibition with the *B. subtilis* strain compared to the *P. aeruginosa* strain, which can be explained by the structural variations of its cell wall. The former is Gram (+), which has a thick protective peptidoglycan layer that easily facilitates the absorption of Ag^2+^ ions; in contrast, the Gram (-) bacterium *P. aeruginosa,* where the peptidoglycan is immersed in multiple thin layer membranes, has superior resistance to antibacterial agents. These results are in agreement with those of the study conducted by Patil et al. [56]. The inhibitory power of the silver film was observed from each of the used reference strains, showing a high efficacy in the reduction of CFU due to the release of Ag^2+^ ions.

The multilayer coating significantly inhibited the growth of *C. albicans* from the first 48 h of incubation until 96 h compared to that obtained with *C. krusei,* which presented intrinsic resistance to fluconazole due to the reduction of the susceptibility of its C-14 α-demethylase. Thus, the synthesis of ergosterol in the cell membrane was not completely interrupted, unlike with *C. albicans*, where ergosterol synthesis was interrupted, thus generating high permeability and stopping fungal growth [57,58]. Pat et al. [59] reported that Ag nanoparticles cause the formation of pores in the membrane and cell wall in *C. albicans,* thus blocking the budding process and possible cell death, which confirms the decrease in CFU when compared to *C. krusei.*
Table 4 shows the analysis of variance of the adherence of the studied strains to the coatings.

Table 5 shows the inhibitory power of silver in each of the monolayer and multilayer coatings (as calculated with the analysis of variance), using the Ti coating as a control. As the number of Ag layers increased, the growth of the different strains of bacteria and yeasts, except for *C. krusei*, decreased; these results were statistically significant.

The formation of *Candida* spp. biofilms releases planktonic cells into the bloodstream, which is lethal in immunocompromised patients or those weakened by surgery; an alarming factor is the formation of mixed polymicrobial biofilms (fungal/bacterial) that present resistance to antifungal agents. In addition, it has been demonstrated that *Staphylococcus epidermidis* protects *C. albicans* against fluconazole and amphotericin B. [28].

Though generalized argyria is unlikely to occur at respirable silver concentrations in air (0.01 mg.m^−3^) or cumulative oral doses of less than 3.8 g [60], Ag nanoparticles (AgNPs) and AgNO_3_ can alter the specification of the endoderm and mesoderm [61]. The differentiation of the processes suggests that the toxicity of AgNPs may not be exclusively due to the release of silver ions, but it is necessary to carry out toxicological evaluations in future biomedical applications.

## 3. Materials and Methods

### 3.1. Materials

Ethanol, formamide, phosphoric acid, sodium hydroxide, glucose, CASO broth, bacteriological peptone, and trypticase agar were supplied by Merck (Kenilworth, NJ, USA). Acetone (J.T. Baker, Landsmeer, The Netherlands), propylene carbonate (Sigma-Aldrich, St. Louis, MO, USA), glycerin (Fisher Scientific, Waltham, MA, USA), and Hartmann’s solution or Ringer’s lactate USP (Baxter, Deerfield, IL, USA) were used without further purification. The bacteria *Pseudomonas aeruginosa* (ATCC 15442) and *Bacillus subtilis* (ATCC 6051) and the yeasts *Candida krusei* (ATCC 1789) and *Candida albicans* (ATCC 66027) were evaluated for their activity. Sabouraud agar [1% bacteriological peptone (Oxoid™ Catalog number: LP0037B), 2% D+glucose (108337 Millipore), and 2% bacteriological agar (Sigma-Aldrich, CAS: 9002-18-0)] and Sabouraud broth (1% bacteriological peptone and 2% glucose) were used for analyses.

### 3.2. Deposition of the Films by Magnetron Sputtering

The 3.0 mm thick glass substrates were initially cleaned with deionized water and neutral soap to remove contaminants and grease, and then cleaned with ethanol and acetone in ultrasound for 15 minutes for each treatment. The substrate was sintered at 120 °C. We used Ti and Ag powder targets located at the cathode of the system, and the substrate was located at the anode. The parameters used in the production of 90 nm coatings per layer in argon atmosphere are summarized in Table 6.

### 3.3. Characterization of the Films

#### 3.3.1. Morphological Analysis and X-ray Diffraction

The morphological characterization of the films was performed by SEM (EM-30AX Coxem with a magnification capacity of 100,000X and a resolution of 17 nm per pixel). The verification of the composition of the coatings was performed by grazing X-ray diffraction (Panalytical, Aeris, type 207055), with continuous scanning in the range between 20° ≤ 2θ ≤ 90°; a Cu anode; K-alpha1 [Å] 1.544060, K-alpha2 [Å] 1.544443, and K-beta [Å] 1.39225 detector lines; a Ka1/Ka2 ratio of 0.5; a power of 15 mA; and flow rate of 40 kV.

#### 3.3.2. Wettability Studies

Penetrability was determined by the contact angle generated between the film and different solvents (propylene carbonate, glycerin, formamide, Hartmann’s solution, or Ringer’s lactate USP and distilled water) using a video system (digital microscope from Levenhuk DTX 50, software included). Measurements were performed at room temperature (T = 25 °C). Droplets of 20 µL were deposited onto the coatings with a micrometer syringe while implementing the sessile drop technique. Contact angle readings were evaluated upon drop deposition and after 120 s of deformation. The above-mentioned procedures were performed in triplicate, with new surfaces available for each film.

#### 3.3.3. Corrosion of the Films

The analyses were performed in a potentiostat (Autolab B.V Metrohm), with a single cell consisting of a working electrode (representing the coatings) evaluated with an exposed area of approximately 6.25 mm^2^ of the conductive film, an Ag/AgCl reference electrode, and a platinum wire counter electrode in a solution of serum Ringer’s lactate USP, Hartmann’ isotonic medical solution simulated the mineral concentration of the physiological fluids of blood, as shown in Table 7. Tafel polarization measurements were obtained at room temperature. The open circuit potential (OCP) was established by leaving the electrodes submerged for one hour before performing ±0.2 V runs.

### 3.4. Antimicrobial Activity of the Films

The antimicrobial activity of the bacteria *P. aeruginosa, B. subtilis, C. krusei,* and *C. albicans* was evaluated. The protocol proposed by Unosson [10], with modifications, was followed. Initially, an OD_600_ absorbance of 0.2, corresponding to ~10^8^ CFU, was established in a UV–visible spectrophotometer (Spectroquant^®^ Prove 600). Bacteria were inoculated in CASO broth and fungi in Sabouraud broth. All broths and agars were prepared with Milli-Q water- The CFU quantification of the strains extract exposed to the films was performed after dilutions until readings between 30 and 100 CFU were achieved.

Autoclaved 6.25 cm^2^ films (BIOBASE), sterilized at 134 °C and 0.22 MPa for one hour, were inoculated in duplicate with 1.0 mL of the strains and then incubated (Memmert BE 400) for two hours at 37 °C. Subsequently, the films were washed with 2.0 mL of phosphate buffer at pH 7.4 and 0.2 M ionic strength to remove unattached microorganisms. Next, the films were sonicated in 5.0 mL of buffer for 120 s at room temperature in order to separate and suspend the adhered bacteria, and then 1.0 mL of the suspension was taken and diluted in broth. Next, 0.1 mL of broth was uniformly inoculated in soy trypticase agar after 24 h of incubation at 37 °C for bacteria and after 48 h of incubation at room temperature for fungi. Finally, CFU data were subjected to an analysis of variance using Tukey’s test (InfoStat/L, version 2020) with a significance level of <0.05.

## 4. Conclusions

Our wettability studies evidenced a decrease in the contact angle over an interval of 120 s due to deformation until reaching equilibrium, thus conferring high efficiency in the cell adhesion of Ag-terminated coatings. The multilayer coating with the highest penetrability, a high surface energy, and a good adhesiveness turned out to be the Ti/Ag film. Structural characterization using X-ray diffraction and the grazing beam technique revealed the presence of TiO_2_ in the anatase crystalline phase, which is a metastable phase that can undergo an irreversible phase transition to rutile which exhibits higher stability, Ag was arranged in a face-centered cubic lattice, and titanium presented an almost amorphous structure. SEM morphological analysis indicated the presence of porous cores in the Ag coating and crystals embedded on the surface of the Ti film.

The Ti/Ag and Ag/Ti/Ag/Ag/Ti/Ag/Ti/Ag/Ti coatings showed promise for applications in biomedical devices by presenting excellent bacterial inhibition, excellent fungal inhibition, and good anticorrosive properties. The multilayer coating presented certain incompatibilities despite having demonstrated a stronger ability to form TiO_2_ layers with a high protective capacity against corrosion and higher CFU inhibition when evaluated against *C. krusei* and *C. albicans* fungi. The SEM of the film showed that three layers on the surface demonstrated detachment, a factor that can put the useful life of a biomedical device at risk when implanted in the host organism. The Ti/Ag coating showed excellent penetrability, biocompatibility, and antibacterial properties, thus making it an alternative implant material in the field of biomaterials.

## Figures and Tables

**Figure 1 molecules-26-04813-f001:**
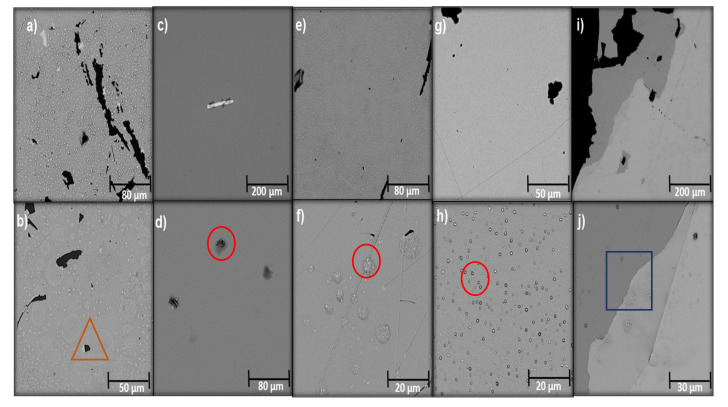
SEM of the deposited films: (**a**) Ag, (**b**) Ag magnification, (**c**) Ti, (**d**) Ti magnification, (**e**) Ti/Ag, (**f**) Ti/Ag magnification, (**g**) Ti/Ag/Ti/Ag, (**h**) Ti/Ag/Ti/Ag magnification, (**i**) Ag/Ti/Ag/Ti/Ag/Ti, and (**j**) Ag/Ti/Ag/Ti/Ag/Ti magnification. Triangle: holes; circles: embedded crystals; square: bonding interface of Ti and Ag.

**Figure 2 molecules-26-04813-f002:**
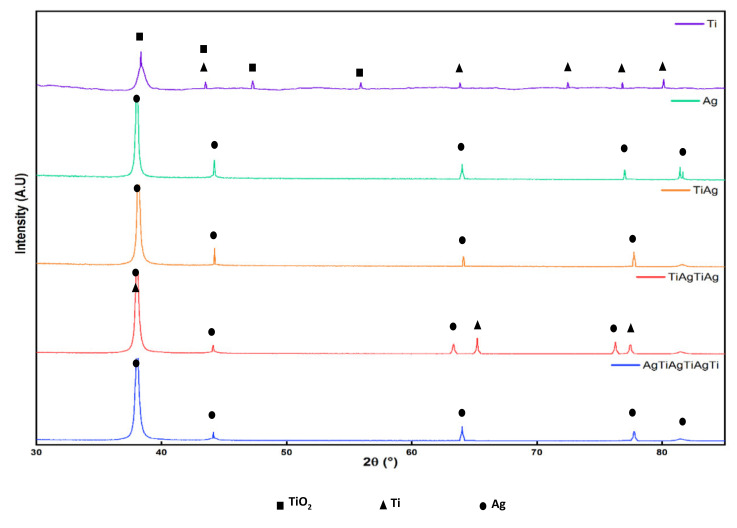
Diffractograms obtained for Ti and Ag monolayer and multilayer coatings deposited on glass substrates.

**Figure 3 molecules-26-04813-f003:**
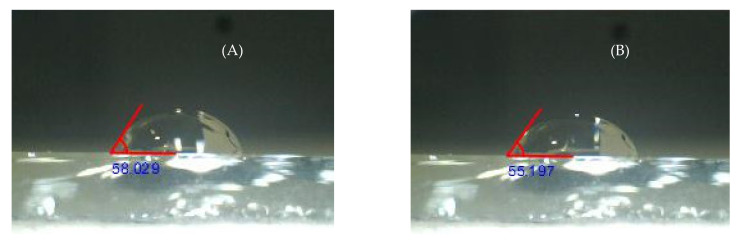
Contact angle of water droplet with 20 µL of volume on Ti/Ag/Ti/Ag film (**A**) t = 0 s and (**B**) t = 120 s.

**Figure 4 molecules-26-04813-f004:**
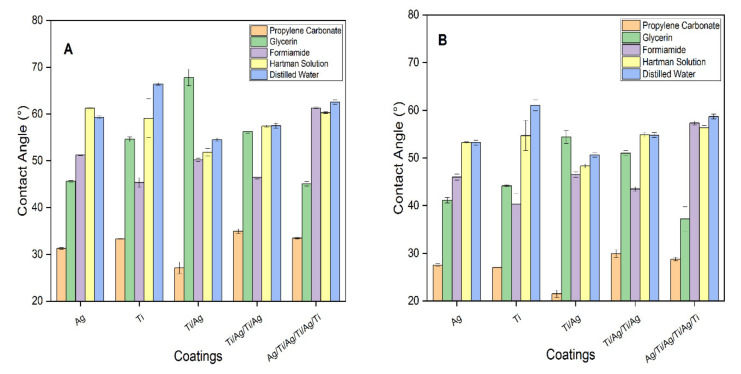
Contact angle variation: (**A**) t = 0 s and (**B**) t = 120 s.

**Figure 5 molecules-26-04813-f005:**
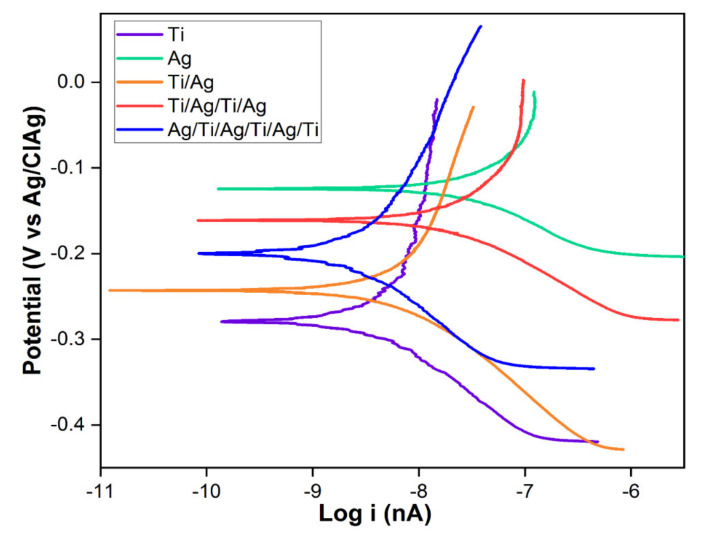
Tafel polarization curves of the studied coatings during 1h of continuous exposure in Hartmann’s solution or Ringer’s lactate.

**Figure 6 molecules-26-04813-f006:**
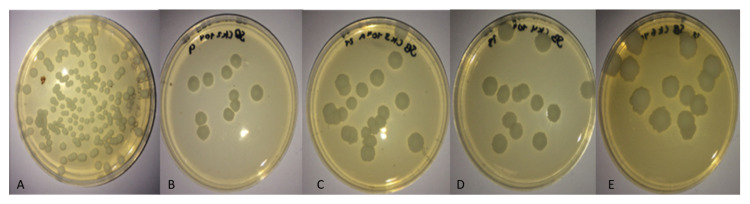
Colonies of *C. krusei* after 96 h of incubation at room temperature: (**A**) inoculum, (**B**) Ag, (**C**) Ti, (**D**) Ti/Ag, and (**E**) Ag/Ti/Ag/Ti/Ag/Ag/Ti.

**Table 1 molecules-26-04813-t001:** Surface energy results calculated with the Young–Dupre equation.

Film	Surface Energy (mJ.m^2^)				
	Propylene Carbonate	Glycerin	Formamide	Hartmann’s Solution	Water
Ag	78.091	100.950	94.825	89.178	109.964
Ti	77.308	93.804	99.273	91.116	101.905
Ti/Ag	79.577	81.824	95.635	97.405	115.075
Ti/Ag/Ti/Ag	76.601	92.420	98.428	92.614	111.908
Ag/Ti/Ag/Ti/Ag/Ti	77.228	101.342	86.296	90.032	106.383

γ_LV_ at 25 °C (mJ.m^2^): propylene carbonate 42.1; glycerin: 59.4; formamide: 58.3; water: 72.8; Hartmann’s solution: 60.2.

**Table 2 molecules-26-04813-t002:** Tafel polarization results.

Coating	Tafel Extrapolation		
	E_corr_ (V)	I_corr_ (nA)	J_corr_ (nA·cm^−2^)
Ti	−0.2791	4.639	0.2899
Ag	−0.1239	32.70	2.044
Ti/Ag	−0.2434	7.684	0.4802
Ti/Ag/Ti/Ag	−0.1609	22.38	1.399
Ag/Ti/Ag/Ti/Ag/Ti	−0.2002	2.798	0.1749

**Table 3 molecules-26-04813-t003:** Microbiological adherence in the coatings.

Coating	Adherence Percentage (%)					
	*Pseudomonas aeruginosa*	*Bacilus subtilis*	*Cándida krusei*		*Cándida albicans*	
			48 h	96 h	48 h	96 h
Glass	0.11	8.33	2.39	2.45	0.41	0.44
Ag	2.57	16.67	2.49	2.60	0.73	1.13
Ti	5.24	27.08	4.23	4.31	2.30	2.22
Ti/Ag	0.17	1.08	3.48	3.43	0.32	0.44
Ti/Ag/Ti/Ag	1.58	5.25	6.12	6.03	3.31	3.35
Ag/Ti/Ag/Ti/Ag/Ti	1.27	1.33	3.13	3.19	0.16	0.15

**Table 4 molecules-26-04813-t004:** Analysis of variance of the films on each of the strains using Tukey’s test.

Strain		Coating						*p*-Tukey
		Glass	Ti	Ag	Ti/Ag	Ti/Ag/Ti/Ag	Ag/Ti/Ag/Ti/Ag/Ti	
*C. krusei*	48 h	X		X	X	X	X	0.0091
	96 h	X		X		X		0.0183
*C. albicans*	48 h	X	X	X	X	X	X	0.0459
	96 h					X	X	0.0348
*B. subtilis*			X		X	X	X	0.0004
*P. aeruginosa*		X	X	X	X	X	X	<0.0001

X: Coatings in which they differ significantly for the same microorganism, according to Tukey’s test.

**Table 5 molecules-26-04813-t005:** Analysis of variances when evaluating the inhibitory power of Ag in the coatings.

Coating	Strain					
	*Cándida krusei*		*Cándida albicans*		*Bacilus subtilis*	*Pseudomonas* *aeruginosa*
	48h	96h	48h	96h		
Ag						0.0158
Ti/Ag			0.0434		0.0064	0.0003
Ti/Ag/Ti/Ag					0.009	0.0016
Ag/Ti/Ag/Ti/Ag/Ti			0.0326	0.0198	0.0067	0.0013

**Table 6 molecules-26-04813-t006:** Film deposition conditions used in magnetron sputtering.

Film	Power (KW)	Pressure (×10^−3^) mBar
Ag	10.9	4.8
Ti	56	2.1
Ti/Ag	56/11.5	2.1/5
Ti/Ag/Ti/Ag	56/11.5/34/15	2.1/5/3.3/4.9
Ag/Ti/Ag/Ti/Ag/Ti	10.9/56/11.5/34/15/40.2	4.8/2.1/5/3.3/4.9/2.3

**Table 7 molecules-26-04813-t007:** Comparison of ionic concentration in blood plasma and Hartmann’s solution.

Ions	Na^+^	K^+^	Ca^2+^	Cl^−^	pH
Blood plasma	142.0	5.0	2.5	103.0	7.2–7.4
Ringer’s lactate	131.0	5.0	4.0	111.0	6.0–7.5

## Data Availability

Not applicable.

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
