# Peer review of "Study of Titanium–Silver Monolayer and Multilayer Films for Protective Applications in Biomedical Devices"

_molecules, 2021, doi:10.3390/molecules26164813_

Round 1
Reviewer 1 Report
The manuscript presented for evaluation, titled 'Study of titanium-silver monolayer...' and submitted by S-P Castro-Narvaez, has as main topic the study of Ti/Ag films in different configurations, in order to improve the protective applications for biomedical devices. The manuscript is well written and contains 54 References, most in the correct format ( for example, for the last one the name of the journal is not written in italics). Besides, sometimes references are written like [19][20][21], while is better to be as [19-21].
Characterization of the film were performed by X-ray diffraction, SEM, wettability studies, corrosion and by the antimicrobial activity of them. The film were obtained by deposition using magnetron sputtering. The details of the science involved are good enough to make it reproducible. The conclusions are adequate and took into consideration all the discussion and the experimental results. The multilayer Ti/Ag film seems to be the one with highest penetrability and good adhesiveness. These coatings showed very good properties for further applications as biomedical devices due to the fact that they are presenting excellent bacterial inhibition properties, towards fungal biofilms. The anticorrosive properties are also a good asset regarding such materials.
In conclusion, the paper can be published as it is.
Author Response
The authors thank you for your appreciation.
Response to concerns:
Point 1. "The manuscript presented for evaluation, titled 'Study of titanium-silver monolayer...' and submitted by S-P Castro-Narvaez, has as main topic the study of Ti/Ag films in different configurations, in order to improve the protective applications for biomedical devices. The manuscript is well written and contains 54 References, most in the correct format ( for example, for the last one the name of the journal is not written in italics). Besides, sometimes references are written like [19][20][21], while is better to be as [19-21]"
Response 1. The references were organized according to the suggestions.
Reviewer 2 Report
Comments:
- Improving antibacterial activity can inhibit infection. We found some researches on silver incorporation to achieve antibacterial effect about surface modification in biomaterials. Thus, it will be better if clarify the interaction between antibacterial performance and bone regeneration into 1. Introduction. Provide these strategies to improve infection through increasing antibacterial rate and incorporating the active ingredients.
- Liu, Q. Wang, W. Liu, et al. Multi-scale Hybrid Modified Coatings on Titanium Implants for Non-cytotoxicity and Antibacterial Properties, Nanoscale, 2021.
- Wang, P. Zhou, S. Liu, et al. Multi-Scale Surface Treatments of Titanium Implants for Rapid Osseointegration: A Review , Nanomaterials, 2020, 10: 1244.
Yang Z, Gu H, Sha G, Lu W, Yu W, Zhang W, Fu Y, Wang K, Wang L. TC4/Ag Metal Matrix Nanocomposites Modified by Friction Stir Processing: Surface Characterization, Antibacterial Property, and Cytotoxicity in Vitro. ACS Appl Mater Interfaces 2018, 10(48): 41155-41166.
Jin G , Qin H , Cao H , et al. Synergistic effects of dual Zn/Ag ion implantation in osteogenic activity and antibacterial ability of titanium. Biomaterials, 2014, 35(27):7699-7713.
Hengel I , Gelderman F , Athanasiadis S , et al. Functionality-packed additively manufactured porous titanium implants. Materials Today Bio, 2020, 7.
- Please mark the crystals, holes and the bonding interface of Ti and Ag in Figure 1.
- Please mark the phase composition in Figure 2.
- Please explain that the reason of corrosion resistance in different samples is differentin 1.2.
- In 3.2, please investigate the toxic dose of Ag in other articles.
Author Response
The authors welcome your comments that allow us to improve the written material.
We attach the final version of the brief.
Point 1.
"Introduction. Provide these strategies to improve infection through increasing antibacterial rate and incorporating the active ingredients
Liu, Q. Wang, W. Liu, et al. Multi-scale Hybrid Modified Coatings on Titanium Implants for Non-cytotoxicity and Antibacterial Properties, Nanoscale, 2021
Wang, P. Zhou, S. Liu, et al. Multi-Scale Surface Treatments of Titanium Implants for Rapid Osseointegration: A Review , Nanomaterials, 2020, 10: 1244
Yang Z, Gu H, Sha G, Lu W, Yu W, Zhang W, Fu Y, Wang K, Wang L. TC4/Ag Metal Matrix Nanocomposites Modified by Friction Stir Processing: Surface Characterization, Antibacterial Property, and Cytotoxicity in Vitro. ACS Appl Mater Interfaces 2018, 10(48): 41155-41166.
Jin G , Qin H , Cao H , et al. Synergistic effects of dual Zn/Ag ion implantation in osteogenic activity and antibacterial ability of titanium. Biomaterials, 2014, 35(27):7699-7713.
Hengel I , Gelderman F , Athanasiadis S , et al. Functionality-packed additively manufactured porous titanium implants. Materials Today Bio, 2020, 7."
Response 1:
The following paragraph was included in the introduction:
"Strategies aimed at the form of incorporation of antibacterial ingredient have been the subject of study, hybrid coatings, micro- and nanoscale modification and biomimetic functionalization of titanium surfaces show rapid osseointegration [29, 30].
The implantation of dual Zn/Ag ions in osteogenic activity presents synergistic effects on the antibacterial capacity of titanium due to long-range interactions generated by Zn and short-range interactions of Ag derived from the microgalvanic pairs in the coimplanted titanium [31].
The use of Ag nanocomposites modified by friction stir processing establishes an antibacterial effect independent of the release of Ag ions, but independent of the number of silver nanoparticles embedded on the surface,
which directly contact and subsequently destroy the bacterial cell membrane without showing any cytotoxicity to bone mesenchymal stem cells in vitro [32]. Likewise, the use of porous titanium implants electrolytic oxidation of plasma with strontium and silver nanoparticles show excellent surface biofunctionalization [33]"
Point 2: Please mark the crystals, holes and the bonding interface of Ti and Ag in Figure 1.
Response 2: In the figure, geometric structures were added for their differentiation
Point 3: Please mark the phase composition in Figure 2. Response 3: In the figure, geometric structures were added for their differentiation Point 4: Please explain that the reason of corrosion resistance in different samples is differentin 1.2
Response 4: Appears in the text of the article:
"The extrapolation of the Tafel curves shows that the multilayer coatings present good corrosion resistance
with respect to the Ag coating, which indicates that the film promotes the attack of the solution in which it is
immersed, due to the degree of porosities formed on its surface, these can be produced in the coating due to
nucleation phenomena as reported by the authors Correa et al [34], being Ti which confers an increase of this property
in the coatings of these alloyed metals thanks to an increase of the microstructural homogenization and by presenting a crystalline phase observed by XRD analysis which is related to previous research by Kumari and Majumdar [53].
In the multilayer coatings, the intermetallic character can be appreciated by obtaining intermediate values, where
Ag contributes an increase in the corrosion current (icorr) of the multilayer films thanks to its ease to experience
corrosion and to being a film that presents little homogeneity or pores on its surface"
Point 5: In 3.2, please investigate the toxic dose of Ag in other articles.
Response 5: A final paragraph was added in the discussion of results with the requested information " Although generalized argyria is unlikely to occur at respirable silver concentrations in air of 0.01 mg*m-3
or cumulative oral doses less than 3.8g [60], Ag nanoparticles (AgNPs) and AgNO3 alter the specification of the endoderm and mesoderm [61].
The differentiation of the processes estimates that the toxicity of AgNPs may not be exclusively due to the release of silver ions,
but it is necessary to carry out toxicological evaluations in future biomedical applications."
Gratefully
Reviewer 3 Report
This paper provides information on the surface energy of Ti and Ag thin films in a different layer and multilayer configurations obtained by cation bombardment sputtering coating, complemented with microbiological analysis of compatibility and corrosion in Ringer's lactate salt solution stress. The paper is well written, but some typos were found, especially in the references, where duplicate references were found. The subject to be studied is very well presented in the introduction, with current and adequate references (around 79% are in the last five years). The methods were clearly presented and were the classical methodology for preparing and characterizing this class of materials. The results are very interesting, and their analysis is adequate, based on adequate earlier findings. The conclusions are solid, supported by very interesting results. This paper may add very interesting knowledge for titanium containing antimicrobial agents. I suggest its publication after a revision on the grammar and writing.
Author Response
The authors appreciate your appreciation.
Response to concerns:
Point 1."This paper provides information on the surface energy of Ti and Ag thin films in a different layer and multilayer configurations obtained by cation bombardment sputtering coating, complemented with microbiological analysis of compatibility and corrosion in Ringer's lactate salt solution stress. The paper is well written, but some typos were found, especially in the references, where duplicate references were found. The subject to be studied is very well presented in the introduction, with current and adequate references (around 79% are in the last five years)".
Response 1. The organization of references in alphabetical order, does not establish repetition of citations (see attached document). The overcitation in the text was accommodated to improve the writing.
Point 2. "The methods were clearly presented and were the classical methodology for preparing and characterizing this class of materials. The results are very interesting, and their analysis is adequate, based on adequate earlier findings. The conclusions are solid, supported by very interesting results. This paper may add very interesting knowledge for titanium containing antimicrobial agents. I suggest its publication after a revision on the grammar and writing".
Response 2. A grammatical revision of the writing was carried out.
